# Drivers of linkage disequilibrium across a species' geographic range

Kay Lucek ⬤ *, Yvonne Willi

Department of Environmental Sciences, University of Basel, Basel, Switzerland

* kay.lucek@unibas.ch

## Abstract

While linkage disequilibrium (LD) is an important parameter in genetics and evolutionary biology, the drivers of LD remain elusive. Using whole-genome sequences from across a species' range, we assessed the impact of demographic history and mating system on LD. Both range expansion and a shift from outcrossing to selfing in North American *Arabidopsis lyrata* were associated with increased average genome-wide LD. Our results indicate that range expansion increases short-distance LD at the farthest range edges by about the same amount as a shift to selfing. However, the extent over which LD in genic regions unfolds was shorter for range expansion compared to selfing. Linkage among putatively neutral variants and between neutral and deleterious variants increased to a similar degree with range expansion, providing support that genome-wide LD was positively associated with mutational load. As a consequence, LD combined with mutational load may decelerate range expansions and set range limits. Finally, a small number of genes were identified as LD outliers, suggesting that they experience selection by either of the two demographic processes. These included genes involved in flowering and photoperiod for range expansion, and the self-incompatibility locus for mating system.

## Author summary

Nearby genomic variants are often co-inherited because of limited recombination. The extent of non-random association of alleles at different loci is called linkage disequilibrium (LD) and is commonly used in genomic analyses, for example to detect regions under selection or to determine effective population size. Here we reversed testing and addressed how demographic history may affect LD within a species. Using genomic data from more than a thousand individuals of North American *Arabidopsis lyrata* from across the entire species' range, we quantified the effect of postglacial range expansion and a shift in mating system from outcrossing to selfing on LD. We show that both factors lead to increased LD, and that the maximal effect of range expansion is comparable with a shift in mating system to selfing. Heightened LD involves deleterious mutations, and therefore, LD can also serve as an indicator of mutation accumulation. Furthermore, we provide evidence that some genes experienced stronger increases in LD possibly due to selection associated with the two demographic changes. Our results provide a novel and broad view

**Data Availability Statement:** All genomic data is deposited on NCBI: PRJEB19338 (pool-seq; https://www.ncbi.nlm.nih.gov/bioproject/?term=PRJEB19338), PRJEB30473 (individual sequences; https://www.ncbi.nlm.nih.gov/bioproject/?term=PRJEB30473). All used LD

estimates from pool-seq and individual genomes are deposited on Zenodo 10.5281/zenodo.4537482 (https://zenodo.org/record/4537482).

**Funding:** This study was funded by the Swiss National Science Foundation, Grant Number: PP00P3_123396, PP00P3_146342, 31003A_140979 and 31003A_166322 to YW; and Fondation Pierre Mercier pour la Science, Lausanne to YW. The funders had no role in study design, data collection and analysis, decision to publish, or preparation of the manuscript.

**Competing interests:** The authors have declared that no competing interests exist.

on the evolutionary factors shaping LD that may also apply to the very many species that underwent postglacial range expansion.

## Introduction

Linkage disequilibrium (LD) measures the non-random coinheritance of alleles at different loci [1]. LD is a common operational variable in genetics and evolutionary biology [2], used among others to map quantitative trait loci [3], estimate effective population size and past founder events [4,5], or to detect genomic regions under selection[6]. However, LD does not unequivocally reveal underlying genomic and evolutionary processes because it can be influenced simultaneously by several factors [2]. For example, theory predicts that average LD increases across the genome due to demographic bottlenecks and enhanced genetic drift [7,8], and locally in some genomic regions, as a result of selection or structural genomic rearrangements such as inversions [9]. Conversely, gene flow and recombination are predicted to reduce LD [2]. Because multiple factors can shape LD, identifying the relevant evolutionary processes is challenging and requires the study of replicate populations at a genomic and genic level. Here we focused on the role of two demographic factors that are predicted to foremost affect genetic drift and as a consequence LD. Based on whole-genome sequences of many replicate populations of North American *Arabidopsis lyrata*, we assessed the role of range expansion and mating system shift in shaping current patterns of LD across an entire species' range.

Range expansion generally happens via serial founder events along the expansion axis and is accompanied by heightened genetic drift [10]. One of the consequences of genetic drift is increased LD [2,7,11] that may extend over larger genomic regions [8]. Average genome-wide LD is predicted to increase because founder events result in the loss of some ancestrally recombined haplotypes and the increase of others [12,13]. However, due to the stochastic nature of the process of genetic drift, the magnitude of LD is expected to vary greatly across the genome [11,14]. Drift-induced LD is predicted to constrain the response to directional selection and the adaptive potential of populations. During range expansion, individuals need to establish, and populations need to frequently adapt to novel environmental conditions. LD may reduce the potential for local adaption because it slows the spread of favorable alleles at linked loci on their way to fixation [15,16]. Furthermore, LD may also involve deleterious mutations as mutational load is predicted to accumulate via genetic drift under range expansion [17]. The interaction between LD, genetic drift and mutational load may further reduce the efficacy of directional selection [18]. Despite the aforementioned theoretical predictions, empirical evidence for an increase in LD with range expansion is limited to humans. Here the expansion out of Africa was found to be associated with a decline in genetic diversity and an increase in LD with distance from the presumed origin of the species [19,20], while subsequent founder events and bottlenecks further contributed to increased LD in some populations [21]. More research on the magnitude of impact of range expansion on LD is therefore needed, given that most if not all species once expanded their range, and that most species remain having fairly dynamic ranges, which has become most obvious under recent climate change [22].

A shift in mating system from outcrossing to selfing evolved repeatedly in many plant families [23] as well as in some populations within primarily outcrossing species [24,25]. Selfing decreases the drift-effective population size relative to outcrossing, reducing genetic variation [26] and effective recombination [27]. As a consequence, LD is predicted to increase across the genome. While drift due to selfing is a stochastic process causing variation in LD across the genome, the effect of drift is predicted to be lower if selfing occurs more rapidly than the loss of variability from random genetic drift [26]. Extended LD is further predicted to increase the

effect of linked selection, i.e. hitchhiking effects over larger genomic distances in highly selfing species [28,29]. A couple of past studies compared LD between closely related pairs of selfing and outcrossing species, i.e. between *Arabidopsis thaliana* and *A. lyrata* [30,31], and between *Capsella rubella* and *C. grandiflora* [32]. They found that LD was generally higher in the selfing species. A weakness of such comparisons is that species commonly differ in more aspects than just the mating system because their shared history usually lies back up to some million years. Better resolution on the role of mating system is likely achieved by comparing populations within the same species that differ in mating system.

LD may also vary among genomic regions and genes because of selection, be it directional [2,33], balancing [34,35] or background selection against deleterious mutations [36]. Any of these three types of selection can locally increase LD. Indeed, locally increased LD is a common pattern of selective sweeps, whereby a beneficial mutation is swept towards fixation in a population by natural selection [37], together with hitchhiking genomic variants on either side of a mutation [38]. The processes causing extended genomic regions of high LD has been proposed to occasionally lead to linkage between co-adapted genes and eventually cause adaptive divergence among evolutionary lineages and, in the best case, may result in speciation [39].

To study genome-wide patterns of LD in response to range expansion and a shift in mating system from outcrossing to selfing, we analyzed sequence data from North American *Arabidopsis lyrata* subsp. *lyrata* (L.) O'Kane & Al-Shehbaz (hereafter referred to as *A. lyrata*) across its entire geographic distribution in North America [40]. The species is a short-lived perennial, hermaphroditic plant closely related to *A. thaliana* and both genetically and geographically distinct from the European subspecies *A. lyrata* subsp. *petraea*, with no evidence for gene flow between the two subspecies [41]. Following the retreat of the glaciers starting about ~20'000 years ago, *A. lyrata* underwent range expansion from two distinct glacial refugia in North America [42,43] (Fig 1). Along the expansion, and especially at the edges of the current species' distribution, several populations shifted their mating system from predominant self-incompatibility and outcrossing to predominant self-compatibility and selfing, or rarely to mixed-mating [25,42,44]. Both range expansion and a shift to selfing were shown to be associated with a reduction in genomic diversity and an increase in mutational load [43,45].

We estimated near-range LD for both genic and intergenic regions across the genome and at the level of genes for 52 populations using population-based allele frequency data. Furthermore, we estimated the decay of LD over longer sequence ranges based on two individually sequenced diploid genomes per population. Apart from testing for an effect of past range expansion and a shift in mating system on estimates of LD, we considered the impact of gene density. Increased gene density was shown to be negatively correlated with genetic polymorphisms in the closely related plant *A. thaliana*, most likely due to the enhanced action of selection in these regions [46]. Structural genomic changes were not considered because they are thought to be rare in *Arabidopsis* [47,48]. Next, we tested whether LD commonly involved deleterious variants. Analyses were repeated on a population level, by testing for an association between genome-wide LD and within-population genetic variation or an estimate of deleterious counts [43]. Finally, we identified genes whose LD estimates were correlated with range expansion and mating system beyond neutral expectations, to reveal possible candidate genes associated with either of the two demographic processes.

## Material & methods

### Plant material and sampling strategy

Plant material from 52 populations of North American *A. lyrata* covering the entire distribution was collected during the reproductive seasons in 2007, 2011 and 2014 (Fig 1 and S1

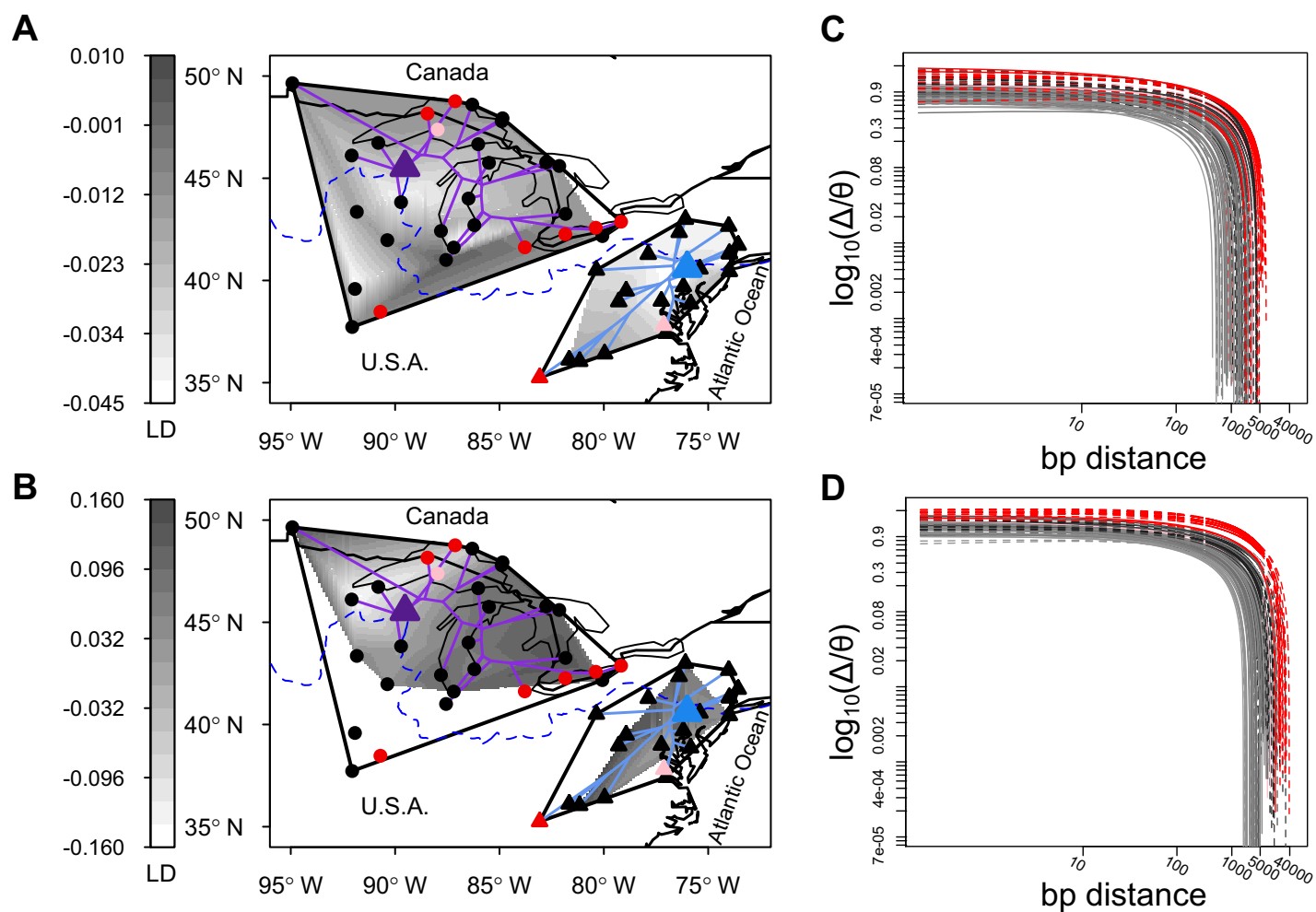

**Fig 1. Geographic pattern of linkage disequilibrium (LD, distance-corrected $r^2$ between pairs of SNPs) in North American *Arabidopsis lyrata* (A&B) and the correlation of zygosity ($\Delta$) scaled by the level of genome-wide heterozygosity ($\theta$) (C&D).** (A) Average genome-wide LD for intergenic regions. (B) Average LD for the gene *FPA* that is involved in the autonomous pathway of flowering time regulation. Darker gray shading for A&B indicates higher levels of LD; both surfaces were estimated based on outcrossing populations only. Minimum convex polygon hulls surrounding populations of the western and eastern ancestral genetic clusters are drawn in black. Triangles indicate the core areas from which recolonization began after the most recent glacial maximum. Lines from these cores are recolonization routes revealed by projecting the population phylogeny on the map. Circles are the populations sampled in this study: outcrossing (black), mixed-mating (pink), and selfing (red). Unshaded areas within the polygon hulls are regions with missing data for outcrossing populations. The dashed blue line indicates the maximum extent of the ice sheet during the last glacial maximum. LD increased with increasing distance from the core areas. The correlation of zygosity ($\Delta$) across intergenic (C) and genic (D) regions for 104 individual genomes as estimated by MLRHO. Solid lines depict individuals from the eastern genetic cluster and dashed lines from the western genetic cluster, respectively. Individuals from mixed-mating populations are depicted in pink, and from selfing populations in red. For individuals from outcrossing populations the grayscale gradient indicates individuals from the range center (gray) to the range edge (black).

Table). Two different genomic datasets were used: first, a previously published dataset of population level whole-genome sequence data (pool-seq) from an equimolar amount of DNA of 25 individuals that was pooled per population before library preparation (NCBI accession PRJEB19338;[43]); second, a dataset consisting of more than a hundred newly re-sequenced genomes with two individuals for each population at ~10X coverage (NCBI accession PRJEB30473). For both datasets, paired-end sequencing was conducted on an Illumina HiSeq 2000 with a read length of 100 bps and 150 bps for pool-seq and the re-sequenced genomes, respectively. Two phylogenetic clusters are distinguished in *A. lyrata*, each with independent recolonizations from different glacial refugia [43]. Our dataset comprised 21 populations from the eastern cluster and 31 populations from the western cluster (Fig 1). While the majority of

these populations were estimated to have established after the withdrawal of the ice, six were found to be older [43]. Two along the Mississippi River in northern Iowa and one in New Jersey were kept in the study because they were close to the deduced glacial refuge areas (Fig 1; S1 Table). Three others were located in Missouri and included one selfing population, apart from two outcrossing populations with clear signatures of past exposure to genetic drift, and they were therefore also kept in the study. For the pool-seq data, all analyses were performed with and without these six populations and results were nearly identical or–for the detection of outlier genes–more conservative when they were included (S1 Text). The mating system of populations was deduced previously by either progeny array analysis or population inbreeding coefficient based on microsatellite markers [42].

## Data preparation and estimation of LD: Pool-seq

With the *trim-fastq.pl* script of POPOOLATION 1.2.2 [49], we first trimmed the raw pool-seq sequences per population using a minimal base quality threshold of 20 and kept only reads $\geq$ 84 bps. We mapped all retained reads against the *A. lyrata* reference genome v1.0 [47] and the plastid and mitochondrial genomes of *A. thaliana* (Genbank accessions NC_000932 and NC_001284, respectively) using BWA-MEM 0.7.13 [50]. The centromeric regions were masked, along with two regions on scaffold II (position ranges: 8'746'475–8'835'273 and 9'128'838–9'212'301) that shared very high similarity with the *A. thaliana* chloroplast genome, suggesting an assembly error in the *A. lyrata* reference genome. Only reads that mapped to scaffolds I-VIII, representing the eight chromosomes of *A. lyrata*, were kept. We next used PICARD 2.1.1 (http://broadinstitute.github.io/picard) to remove duplicate reads and SAMTOOLS 1.7 [51] to retain properly paired reads with mapping quality over 20.

Using SAMTOOLS, we generated a *mpileup* file comprising all populations. SNPs were called using VARSCAN 2.4.1 [52], requiring a minimal read depth of 50 at a given position for each population to make a call. We used a minimal variant allele frequency threshold of 0.03 per population, which is considered a good threshold for pool-seq data [53]. We removed previously identified repeat sites in the *A. lyrata* genome [54] with BEDTOOLS 2.26.0 [55]. Using VCFTOOLS 0.1.14 [56], we then removed indels and kept only biallelic SNP positions that had a coverage of 50–500 within a population, a minimal genotype quality of 28, a minor population allele frequency of 0.03, and a maximum of 50% missing data across all populations. Finally, SNPs with a strand bias higher than 90% were removed. These procedures resulted in 2'586'251 SNPs for downstream analyses.

We calculated LD between SNP positions for each population separately. Because we worked with pool-seq data, the range for estimating LD was short, i.e. across paired reads [57], and ranged between 1 and 500 bps, including the insert size between paired reads. LD was estimated as $r^2$, representing the statistical correlation between pairs of SNPs using the *direct estimate* method of LDX [57], whereby paired reads were considered only if they overlapped at both SNP sites. We considered only sites with a read depth greater than five and a minor allele frequency at either locus > 0.15. The latter has been shown to provide the most accurate estimates of $r^2$ for pool-seq data [57]. Using annotation v1.2 of *A. lyrata* [58], LD estimates were separated into genic and intergenic regions, where genic regions comprised both exons and introns, and intergenic regions covered the sequence between adjacent genes. Genes and intergenic regions were included in the analyses only if they contained at least five LD estimates and <50% of populations had missing data. Of the 32'069 genes in the *A. lyrata* genome [58], 9'282 (29.1%) were retained for all genic-based LD analyses. The same filtering criteria produced LD estimates for 4'972 intergenic regions. The filtered datasets comprised a total of 46'159'344 and 18'801'370 pairwise $r^2$ estimates for the genic and intergenic regions,

respectively, across all populations. Because LD generally decayed with increasing distance between loci (S1 Fig), we calculated the residuals of a linear model between LD and the $\log_{10}$-transformed base pair distance between loci across all retained LD estimates. We subsequently refer to these residuals as distance-corrected LD. Watterson's $\theta$ was estimated with NPSTAT [59] for genic and intergenic regions separately.

We further assessed whether linked SNPs within genic regions were deleterious. For this, we employed the program SIFT 4G [60], which indicates for each missense variant if it is deleterious or tolerated based on site conservatism when compared to homologous sites of a larger database. We ran SIFT 4G with the suggested parameters [60], using the SNP set from the pool-seq data. From the output, we split the pairwise LD estimates into three categories: LD between two deleterious variants, LD between a deleterious and a tolerated variant, and LD between two tolerated variants. For a total of 14'770'554 $r^2$ estimates, both involved SNPs could be annotated by SIFT. Of these, 200'577, 2'129'990 and 12'514'150 estimates occurred between two deleterious variants, between a deleterious and a tolerated variant or between two tolerated variants, respectively.

## Data preparation and estimation of LD: Individual genomes

For the re-sequenced individual genomes, we trimmed the raw sequences with TRIMMOMATIC 0.36 [61] using a minimal base quality threshold of 20 and kept only reads $\geq$ 50bp. We mapped all retained reads against the *A. lyrata* reference genome and removed duplicates and repeat sites as for the pool-seq dataset. For each individual, we calculated the correlation of zygosity ($\Delta$) across all genic sites and across all intergenic sites used for the pool-seq data with MLRHO 2.9 [62]. The correlation of zygosity measures the strength of correlation between loci across individual genomes and provides an estimate for LD [63]. However, rather than resulting in individual pairwise estimates, MLRHO measures $\Delta$ across all loci for each individual. Following the suggestions of Lynch *et al.* [63], we included only sites with a minimal genotype quality of 28 and a sequence depth between twice and five times the average genome wide depth for each individual. We ran MLRHO in steps of 1 bp along the first 5'000 bps and in steps of 100 bps for a distance of 5-50k bps. We scaled $\Delta$ by the level of heterozygosity ($\theta$) estimated for each genome by MLRHO as $\Delta/\theta \cong r^2$ [63].

## Estimation of genome-wide recombination

We assessed if background recombination across the genome of *A. lyrata* could impact LD. For this, we used the software RELERNN [64]. While classic software tools require many individually sequenced specimens per population to confidently estimate recombination, RELERNN implements a machine learning algorithm that can also handle pool-seq data. We ran RELERNN following the suggested best-practice for pool-seq data [64] with the necessary few specific adjustments for *A. lyrata*: For the initial simulation step, we set a recombination rate of $3.55e^{-9}$ crossovers per base per generation, based on the only empirical estimate in this genus from *A. thaliana* [65]. At this step, RELERNN implements MSPRIME [66] to simulate 100'000 training examples and 1'000 validation and test examples based on the observed SNP data for each population. In a next step, RELERNN uses the simulated dataset to train a recurrent neural network for which we used the standard settings as suggested by [64], applying also a minor population allele frequency of 0.03. With the trained network, the original data is then assessed including a bootstrap approach to estimate the 95% confidence interval. Because the calculation of recombination was computationally prohibitive, we ran RELERNN separately for four populations from the cores of distribution with high genomic diversity (IA1, MD4, NY1, WI1, see below). Importantly, RELERNN estimates recombination within self-set

bins, whose sizes are optimized based on the SNP density of the initial datasets; consequently bins differed in lengths (IA1: 137'500 bps, MD4: 63'100 bps, NY1: 61'100 bps, WI1: 61'400 bps). Therefore, we estimated averages of recombination rate for each population, and genic and intergenic region that we used in our LD analyses. Finally, we estimated the relationship between distance-corrected LD and recombination rates that were estimated for each of the four populations for genic and intergenic regions, respectively, with Pearson product-moment correlations. Because LD and background recombination as estimated here were not highly correlated (see Results), we did not include this variable in our statistical analyses.

## Statistical analyses

**i) Preliminaries.** Three preliminary analyses were performed. First, we tested for a difference in LD between genic and intergenic regions. We applied paired $t$-tests between the average genome-wide distance-corrected $r^2$ or the average correlation of zygosity ($\Delta$) scaled by the level of heterozygosity of genic and intergenic regions across populations. For $\Delta$, we focused on the first 500 bps, which represents the range for which $r^2$ was estimated from pool-seq data.

Because selection may result in extended genomic regions of increased LD [39], we secondly aimed at identifying such regions by using pool-seq data only. For this, we employed a hidden Markov model (HMM) as implemented in [67]. We modeled two states (low vs. high) on LD averaged for each genic and intergenic region for each population (S2 Fig) with the R package *HiddenMarkov*. We then estimated the mean LD for each state and the transition rate matrix among states from the data with a Baum-Welch algorithm [68]. Finally, we used the Viterbi algorithm to predict the most likely sequence of hidden states from the data and estimated parameters.

Third, we estimated the relationship between $r^2$ of pairs of SNPs from pool-seq data and $\Delta$ from individual sequencing. We performed Pearson product-moment correlations between the mean per-population $\log_{10}$-transformed $\Delta$ averaged over the first 500 bps and $r^2$, separately for genic and intergenic regions.

**ii) LD based on pool-seq.** A first main analysis tested for the relationship between LD on the one hand, and expansion history or mating system on the other hand. For this, we implemented linear mixed-effects models with restricted maximum likelihoods (REML) –a widely used statistical approach in genetics and more broadly across the field of biology [69–71], with the *lme4* package [72] in R 3.5.1 [73]. Analysis was done separately for genic and intergenic regions. Dependent variable was distance-corrected $r^2$ estimated from pool-seq data, and the four fixed effects were: phylogenetic cluster, $\log_{10}$-transformed expansion distance, mating system, and gene density. While the former three were predictors on the level of the population, the latter was a predictor on the level of chromosomal subunits. *Phylogenetic cluster* assignment of populations in west or east was based on previous analyses. *Expansion distance* was calculated based on a map-projected population phylogeny constructed with genome-wide SNP frequency data [43]. Core areas were defined as the geographic location of the most recent common ancestor of the first population with a current location that had been covered by ice during the last glaciation period. For every population, expansion distance was calculated either as the sum of great circle distances along the map-projected nodes back to the core or, for the six older populations (S1 Table) as the great circle distance directly to the core, in km. *Mating system* was approximated by the population inbreeding coefficient, $F_{IS}$, calculated by microsatellite analyses based mostly on the same plant material as we used for pool-seq [25,42]. *Gene density* was found to vary across the *A. lyrata* genome (S3 Fig). We approximated gene density by the mean base-pair distance of each gene to its respective up- and downstream neighboring genes. For analyses on intergenic regions, gene density was approximated

by the size of each intergenic region. Random effects included population crossed with scaffold and, at the lowest hierarchical level, gene ID or ID of the intergenic region nested within scaffold. For all models, significances were estimated with a type III Wald $\chi^2$ test. We did not include an interaction between range expansion and mating system because they were highly non-significant (genic: $\chi^2 = 0.11$, $p = 0.741$; intergenic: $\chi^2 < 0.01$, $p = 0.992$).

In a second main analysis, the link between LD and load was assessed for genic regions. The goal was to test whether distance-corrected LD estimates increased with expansion distance or mating system independent of whether SNPs were tolerated or deleterious. The analysis as described above was repeated on genic regions. However, additional fixed effects included the three SNP types (deleterious-deleterious, deleterious-tolerated, tolerated-tolerated) and their interaction with range expansion and mating system. For all linear mixed-effects models, we inspected the residuals visually for normality and homoscedasticity.

Two additional analyses focused on effective population size ($N_e$) or mutational load, and were performed on the level of the population. If genetic drift is an important driver of LD, we would expect LD in genic and intergenic regions to increase with lower genomic estimates of $N_e$ [4]. We used Watterson's $\theta$ of intergenic regions as a proxy for effective population size because in the absence of substantial gene flow (as is the case for *A. lyrata* [25]), only effective population size and mutation rate should determine genomic diversity. We also incorporated a previously published estimate of mutational load, measured as the ratio of the number of non-synonymous to synonymous polymorphic sites adjusted for the average frequency of the allele derived from *A. thaliana* ($P_n f_n / P_s f_s$) [43]. We tested for correlations between average genome-wide distance-corrected LD in genic and intergenic regions and $\theta$ and mutational load using Pearson product-moment correlations.

**iii) LD based on individual genome sequences.** A third main analysis targeted the role of expansion history or mating system on the length over which LD unfolds based on the correlation of zygosity ($\Delta$) estimated across individual genomes. The output of MLRHO is $\Delta$ that is averaged across the genome separately for each bp distance interval between variants. Therefore, we fitted linear mixed-effects models for each bp interval. Response variable was $\log_{10}$-transformed $\Delta$, scaled by the level of heterozygosity of either genic or intergenic regions. Because $\Delta$ is estimated across the entire genome, our model included only three of the four fixed effects used for the pool-seq data: phylogenetic cluster, range expansion and mating system. As we had two individuals per population, we included population as a random effect. *P* values were corrected for multiple testing with a false discovery rate (FDR) correction. To quantify the range of LD, we estimated for each individual the bp distance where the scaled $\log_{10}$-transformed $\Delta$ reached a low value of 0.1.

**iv) Identification of genes impacted by range expansion and mating system shift.** We used distance-corrected near-range LD estimated from pool-seq data to identify genes with a significant relationship with range expansion and mating system shift. Gene-by-gene, we tested whether LD was affected by range expansion or mating system with a linear mixed-effects model that included phylogenetic cluster, $\log_{10}$-transformed expansion distance and mating system as fixed effects. We further included an interaction term between expansion distance and mating system. *P*-values were adjusted for multiple testing using a FDR correction. For each gene, we further assessed if LD increased more than expected by chance during range expansion or mating system shift or in other words, whether the model coefficients for a gene were outside of the respective genome-wide coefficient $\pm$ 2 SE among all genic sites (Table 1). We then identified the gene ontology (GO) terms for the genes that remained significant ($p < 0.05$), using the annotation of [58] and restricting our analysis to biological processes. Finally, because it has been shown that balancing selection may particularly affect some genomic regions during a switch in mating system in *A. lyrata* [74,75], we also compared levels

of Watterson's $\theta$ among the genes that showed a strong effect of range expansion or mating system with the level of $\theta$ averaged across all genic regions using paired $t$-tests.

## Results

### Preliminary analyses

Preliminary analyses showed that first, the average genome-wide estimates of LD were higher for genic than intergenic regions for both distance-corrected $r^2$ (paired $t$-test: $t_{1,51} = 38.45$, $p < 0.001$) and the correlation of zygosity–$\Delta$ ($t_{1,105} = 8.17$, $p < 0.001$). Second, the HMM analysis did not identify any extended region of increased LD in any population. Although our studied regions were spread across the entire genome (S2 Fig), some potential islands of low recombination may not have been included as we focused on regions with minimal number of polymorphic sites. Watterson's $\theta$ among the 9'285 genes included in our study was indeed on average higher than among all other genes (paired $t$-test: $t_{1,51} = 25.62$, $p < 0.001$, average difference of $\theta$: 0.002). Third, LD estimated by average $r^2$ from pool-seq data and LD estimated by average $\Delta$ were significantly correlated (genic regions: $\rho = 0.716$, $t_{1,51} = 7.25$, $p < 0.001$; intergenic regions: $\rho = 0.719$, $t_{1,51} = 7.32$, $p < 0.001$, S4 Fig).

Rates of recombination as estimated from pool-seq data by RELERNN varied across the genome of *A. lyrata*, but decreased in most cases towards the centromeres (S5A Fig). The correlation between distance-corrected LD and recombination within each population was though not strong (genic: range of $\rho$ 0.017–0.076; intergenic: range of $\rho$ 0.017–0.033; S5B Fig).

### Factors driving genome-wide linkage disequilibrium: Pool-seq

Genome-wide distance-corrected LD ($r^2$) estimated from pool-seq data was strongly associated with distance of range expansion and a shift in mating system from outcrossing to selfing for both genic and intergenic portions of the genome (Table 1; Figs 1 and S6 Fig). The model-predicted relative increase in LD between the shortest and longest expansion distance was 509% for genic regions and 78% for intergenic regions. The model-predicted relative increase in LD between the most outbred and the most inbred population was 637% for genic regions, and 130% for intergenic regions. Furthermore, LD declined with lower gene density by 141% for genic regions, for which gene density was estimated as the average distance between neighboring genes. In line, LD declined with lower gene density by 395% for intergenic regions, for

**Table 1. Results of the linear mixed-effects models on genome-wide linkage disequilibrium (LD, distance-corrected $r^2$ between pairs of SNPs) or the correlation of zygosity ($\Delta$) of genic regions and (top) intergenic regions (bottom).** For each fixed effect, the model estimate ($\beta$) as well as the chi-square statistics ($\chi^2$) with the associated $p$ value from type III Wald $\chi^2$ tests are given. For the correlation of zygosity, the results for $\Delta$ at 1000 bps distance are shown (see also S7 Fig). Only results for fixed effects are shown.

| Fixed effect | LD ($r^2$) | | | Correlation of zygosity ($\Delta$) | | |
|---|---|---|---|---|---|---|
| | $\beta \pm SE$ | $\chi^2$ | $P$ | $\beta \pm SE$ | $\chi^2$ | $P$ |
| *Genic regions* | | | | | | |
| Genetic cluster (east) | 0.00595 ± 0.00438 | 1.845 | 0.1744 | 0.06643 ± 0.04334 | 2.349 | 0.1253 |
| Log$_{10}$ expansion distance | 0.03872 ± 0.00947 | 16.719 | <0.0001 | 0.13219 ± 0.04067 | 10.563 | 0.0012 |
| Mating system ($F_{IS}$) | 0.02824 ± 0.00844 | 11.186 | 0.0008 | 0.88161 ± 0.08349 | 111.495 | <0.0001 |
| Log$_{10}$ distance to adjacent genes | -0.01290 ± 0.00104 | 154.101 | <0.0001 | - | - | - |
| *Intergenic regions* | | | | | | |
| Genetic cluster (east) | 0.00844 ± 0.00369 | 5.237 | 0.0221 | 0.06211 ± 0.05054 | 1.510 | 0.2191 |
| Log$_{10}$ expansion distance | 0.02774 ± 0.00797 | 12.108 | 0.0005 | 0.11890 ± 0.04743 | 6.283 | 0.0122 |
| Mating system ($F_{IS}$) | 0.02853 ± 0.00711 | 16.106 | <0.0001 | 0.32472 ± 0.09737 | 11.121 | 0.0008 |
| Log$_{10}$ size intergenic region | -0.01444 ± 0.00169 | 73.401 | <0.0001 | | - | - |

which gene density reflected the size of the respective region. Finally, genetic cluster had only a significant effect for intergenic regions (Table 1).

For distance-corrected $r^2$ estimated between annotated SNPs, LD again significantly increased with expansion distance and mating system, while it declined with decreasing gene densities (Table 2). Importantly, both the type of SNP pairs and the interaction between the types of SNP pairs with expansion distance and mating system were significant (Table 2). The model indicated that LD increased for all three types of SNP pairs towards the range margins (Fig 2A) and with increased inbreeding (Fig 2B), but that the effect was less strong for LD estimates involving two deleterious SNPs. In contrast, LD between pairs of tolerated SNPs increased to a similar degree with expansion distance as it did between a deleterious and a tolerated SNP, but to a lesser degree with mating system. Furthermore, population-level analyses confirmed a link between LD, drift and load. Genome-wide average LD ($r^2$) for both genic and intergenic regions increased with declining Watterson's $\theta$ by comparable magnitudes (genic: $\rho = 0.727$, $t_{1,51} = 7.49$, $p < 0.001$; intergenic: $\rho = 0.744$, $t_{1,51} = 7.86$, $p < 0.001$; Fig 3A). Furthermore, LD for both genic and intergenic regions increased with increasing mutational load ($P_n f_n / P_s f_s$) (genic: $\rho = 0.636$, $t_{1,51} = 5.83$, $p < 0.001$; intergenic: $\rho = 0.637$, $t_{1,51} = 5.84$, $p < 0.001$; Fig 3B).

### Factors driving genome-wide linkage disequilibrium: Individual genomes

The extent over which LD estimated by the scaled correlation of zygosity ($\Delta$) unfolds varied among populations (Fig 1C and 1D): the $\log_{10}$-transformed $\Delta$ reached a value of 0.1 at an average of 4'637 bps (range: 1'168–16'601 bps) for genic regions and at an average of 1'129 bps (range: 259–3'191) for intergenic regions. In line, the range at which $\Delta$ reached 0.1 differed on average by 3'509 bps between genic and intergenic regions (paired $t$-test: $t_{1,105} = 12.85$; 95%CI: 2'967–4'050) and increased with the distance of range expansion and a shift in mating system to selfing (Table 1). Genetic cluster was only significant for the intergenic region (Table 1), where LD extended over longer ranges in the western cluster.

The statistical models run separately for each bp distance interval between variants of genic regions implicated that range expansion affected $\Delta$ consistently over the first 16.0 kb with only four exceptions (Panel B in S7 Fig); a shift in mating system to selfing affected $\Delta$ consistently over the first 23.3 kb (Panel C in S7 Fig). For intergenic regions, range expansion affected $\Delta$ consistently over the first 2.3 kb (Panel E in S7 Fig) and mating system over the first 4.4 kb (Panel F in S7 Fig). While genetic cluster did not significantly affect $\Delta$ for genic regions, it did so for intergenic regions, however, not consistently over extended stretches (Panels A and D in S7 Fig). For example at 1000 bps distance only range expansion and mating system showed a significant effect on $\Delta$ for both genic and intergenic regions (Table 1).

**Table 2. Results of the linear mixed-effects model on genome-wide linkage disequilibrium (LD) using only distance-corrected $r^2$ between pairs of SNPs in genic regions that were annotated by SIFT.** For each fixed effect, the model estimate ($\beta$), as well as the chi-square statistics ($\chi^2$) with the associated $p$ value from a type III Wald $\chi^2$ tests are given. Only results for fixed effects are shown.

| | | LD ($r^2$) | |
|---|---|---|---|
| Fixed effect | $\beta$ ± SE | $\chi^2$ | P |
| Genetic cluster (east) | 0.00591 ± 0.00437 | 1.829 | 0.1762 |
| $\log_{10}$ expansion distance | 0.02739 ± 0.00969 | 17.391 | <0.0001 |
| Mating system ($F_{IS}$) | 0.01779 ± 0.00886 | 12.078 | 0.0005 |
| $\log_{10}$ distance to adjacent genes | -0.01189 ± 0.00113 | 113.989 | <0.0001 |
| SIFT-SNP type | -0.03575 ± 0.00577 | 49.428 | <0.0001 |
| $\log_{10}$ expansion distance * SIFT-SNP type | 0.01231 ± 0.00223 | 31.138 | <0.0001 |
| Mating system ($F_{IS}$) * SIFT-SNP type | 0.01023 ± 0.00283 | 38.577 | <0.0001 |

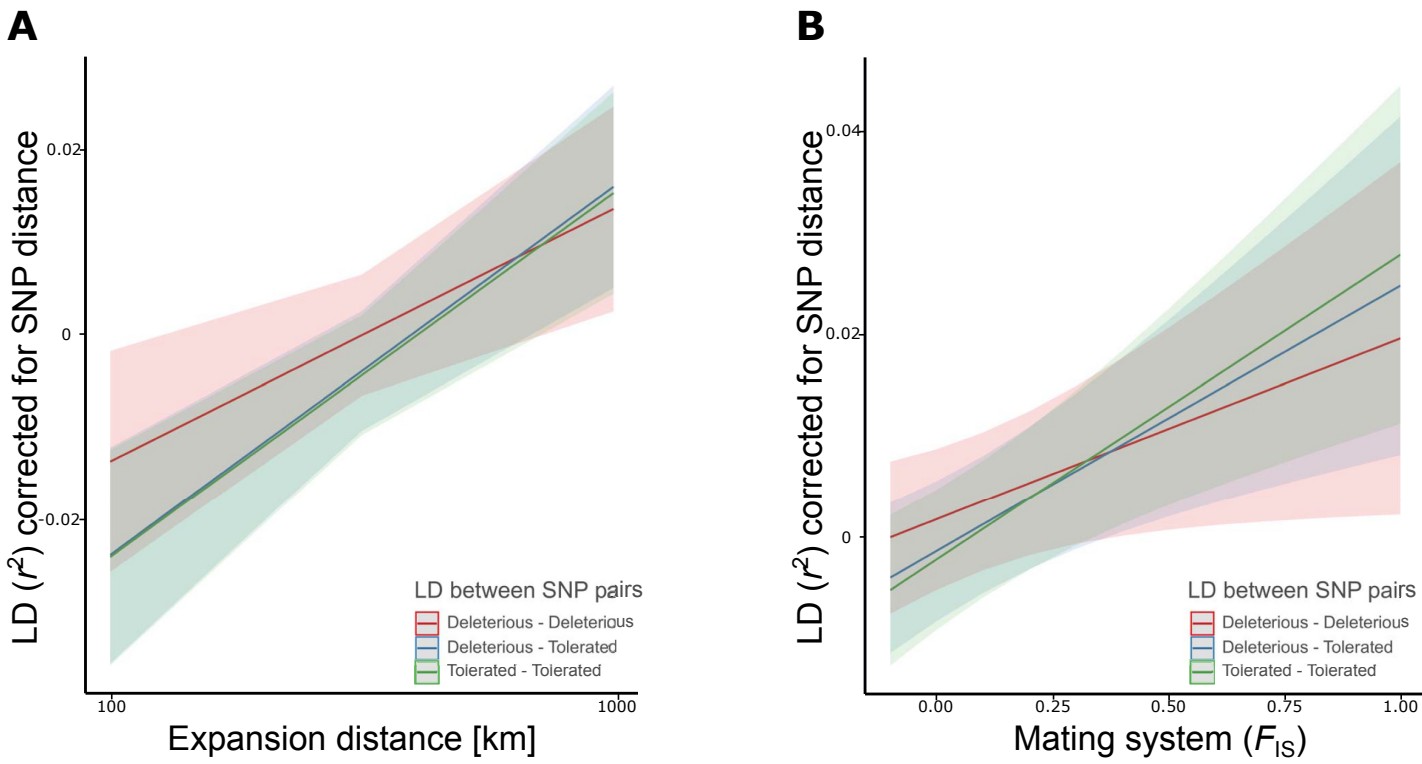

**Fig 2.** Marginal effects for the interactions of LD ($r^2$) estimated from pool-seq between variants that were identified by SIFT to be both deleterious, deleterious and tolerated, or both tolerated, and expansion distance (km, log-scale) (A) and mating system ($F_{IS}$) (B). Regression lines and their 95% confidence intervals were estimated from a linear mixed-effects model (see Table 2).

### Genes likely impacted by range expansion and mating system shift

Gene-specific analyses of distance-corrected LD revealed 271 and 23 genes (2.9% and 0.2% of all studied genes) significantly associated with expansion distance and mating system, respectively, following a false discovery rate (FDR) correction. Only the unannotated gene *AL1G48560* overlapped between the two sets of genes (S2 and S3 Tables), and only for *AL2G23990* a significant interaction between the two predictors was found. A total of 190 and 11 annotated genes were associated with 396 and 42 unique GO terms with range expansion and mating system, respectively, where 26 GO terms overlapped between the two datasets (S2 and S3 Tables). Eight genes and 12 GO terms for expansion distance were linked to *flowering* or *photoperiod*, including the gene *AL4G42150*, a homolog of *FPA* in *A. thaliana* in which it regulates flowering time [76]. Some of the other genes for range expansion were associated with *abiotic stress* (6 genes, 8 GOs), *growth* or *maturation* (5 genes, 12 GOs), *roots* (4 genes, 10 GOs) and *trichomes* (3 genes, 2 GOs). For mating system, one GO term was linked to *recognition of pollen*, being associated with the gene *AL7G32710*, which is the homolog to *ARK3* that flanks the self-incompatibility locus region in *A. lyrata* [77]. While change in LD for *AL7G32710* was strongly associated with the shift in mating system ($\chi^2 = 15.12$, $p < 0.001$; Fig 4), it was not affected by range expansion ($\chi^2 = 0.26$, $p = 0.611$). Further analyses showed that the gene that flanks the self-incompatibility locus on the other side (*AL7G32750* or *Ubox*) was somewhat associated with mating system (Fig 4; $\chi^2 = 5.72$, $p = 0.017$, not significant after FDR) but not with range expansion ($\chi^2 = 0.38$, $p = 0.535$).

Outlier genes for expansion distance had a slight yet significantly lower genetic diversity compared to non-outlier genes ($\Delta_\theta = 0.0002$; paired *t*-test: $t_{1,51} = 14.10$, $p < 0.001$; Fig 5), in

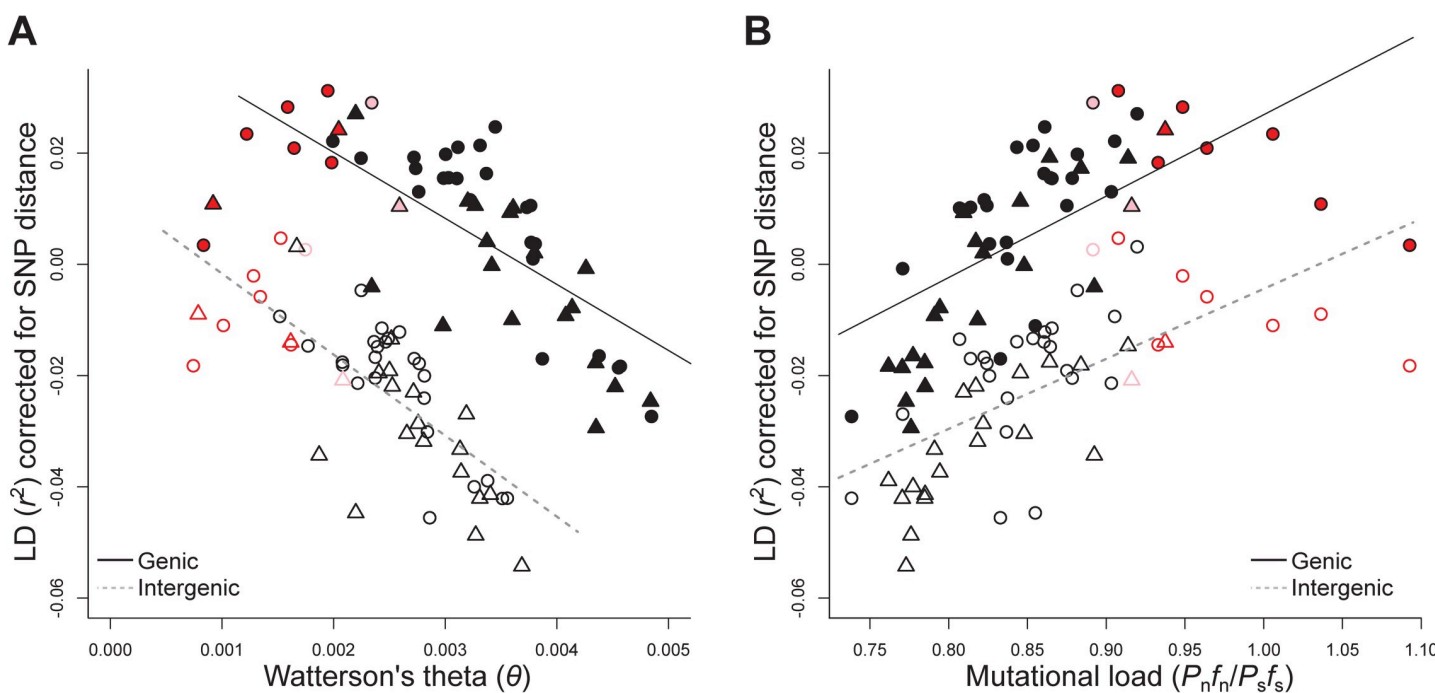

**Fig 3.** Relationship between Watterson's theta ($\theta$) (A), mutational load (B) and average genome-wide linkage disequilibrium (LD, distance-corrected $r^2$ between pairs of SNPs). Mutational load was estimated as the product of non-synonymous polymorphic sites ($P_n$) and their mean derived allele frequency ($f_n$) divided by the analogous product but for synonymous polymorphic sites ($P_s f_s$). Each symbol represents the average estimate of a population. Open and filled symbols are the genomic estimates for intergenic and genic regions, respectively. Circles represent populations from the west, triangles those from the east. Symbol colour indicates the mating system: outcrossing (black), mixed-mating (pink), and selfing (red). The model-predicted regression lines are shown for intergenic (dashed line) and genic (solid line) regions.

line with a potential overall signature of directional selection. In contrast, outlier genes for mating system had a significantly increased $\theta$ compared to non-outlier genes ($\Delta_\theta = 0.0038$; paired $t$-test: $t_{1,51} = 29.30$, $p < 0.001$; Fig 5), consistent with a potential overall signal of balancing selection.

## Discussion

Linkage disequilibrium (LD) is a commonly measured feature of the genome, yet the relative importance of the factors that drive LD are often unknown [2]. Here we showed how patterns of LD on a genome-wide scale are strongly influenced by demographic history, mating system, and gene density (Table 1). Both postglacial expansion distance and a shift to self-fertilization were associated with an increase in LD across the genome, consistent with the increased action of genetic drift, whose effects can be detected over more than 10kb sequence length. The two demographic processes each also affected a distinct set of genes more strongly compared to background increases in LD, suggesting potential targets of selection or linked selection imposed by these demographic processes or factors related to them.

The analyses of LD in this study were based on either short ranges using population-pool sequencing or longer ranges using sequences of individuals of each population. Estimates of LD were used that were most appropriate for the two datasets. For pool-seq, it was the correlation of SNPs ($r^2$), and for individual genomes, it was the correlation of zygosity ($\Delta$) that was scaled by the genome-wide level of heterozygosity. On the one hand, our study clearly showed that the two approaches of estimating LD delivered highly concordant results (S4 Fig). On the other hand, the two types of sequence data provided complementary insights. The sequence

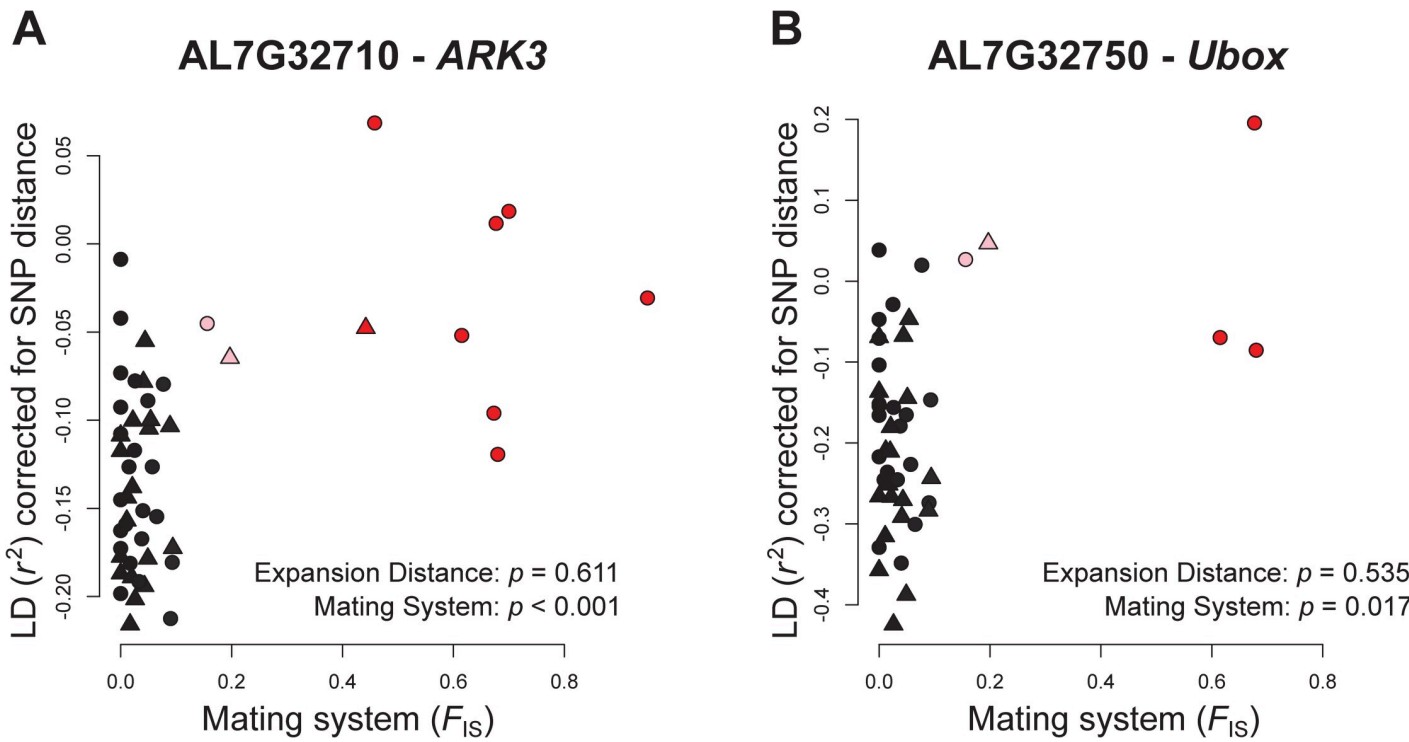

**Fig 4.** Relationship between average linkage disequilibrium (LD, distance-corrected $r^2$ between pairs of SNPs) in the two genes flanking the self-incompatibility locus and mating system, expressed by the population inbreeding coefficient ($F_{IS}$): *AL7G32710-ARK3* (A) and *AL7G32750-Ubox* (B). Each symbol represents the average estimate of a population. The symbol type reflects the genetic clusters (circle for west, triangle for east) and the symbol colour the mating system: outcrossing (black), mixed-mating (pink), and selfing (red). Significance is reported for the effects of expansion distance and mating system, from separate linear models on the data for each gene. For the gene AL7G32750-*Ubox* in B, the relationship with mating system was not significant following a False Discovery Correction.

data of population-pools allowed us to investigate LD over short ranges and increased resolution in analyses, by performing them on the level of genes or sections of intergenic regions. LD estimates from pool-seq were also useful in pinpointing genes putatively experiencing selection. However, multiple short-range LD estimates were required for single genes or intergenic regions, causing some bias for polymorphic regions (Fig 3) and probably the underrepresentation of regions of low recombination. In contrast, the individual-based re-sequence data permitted to estimate the sequence length over which LD extended due to range expansion and the shift in mating system.

Both range expansion and mating system shift increased genome-wide LD (Figs 1 and S4). Populations near the edges of current distributions had been shown to often carry the signature of small population size because of repeated founder events during earlier range expansion [10]. Such founder events are expected to increase LD, as demonstrated for humans with their expansion history out of Africa [19,20] and for one northern population of European *A. lyrata* subsp. *petraea* [78]. Edge populations can also be small and isolated in the absence of recent expansion due to reduced habitat suitability or long-term isolation [79], which may similarly cause higher LD. Consistent with these predictions, LD increased in *A. lyrata* with expansion distance or direct distance to the core of the glacial refugium when using pool-seq data (Figs 1A and S4) and extended over larger sections as shown with individual-level sequences (Figs 1B and 1D). Similar to range expansion, selfing should decrease the drift-effective population size relative to outcrossing [26], reduce effective recombination [27,80] and result in increased LD across the genome [32]. Genome-wide LD in *A. lyrata* was indeed

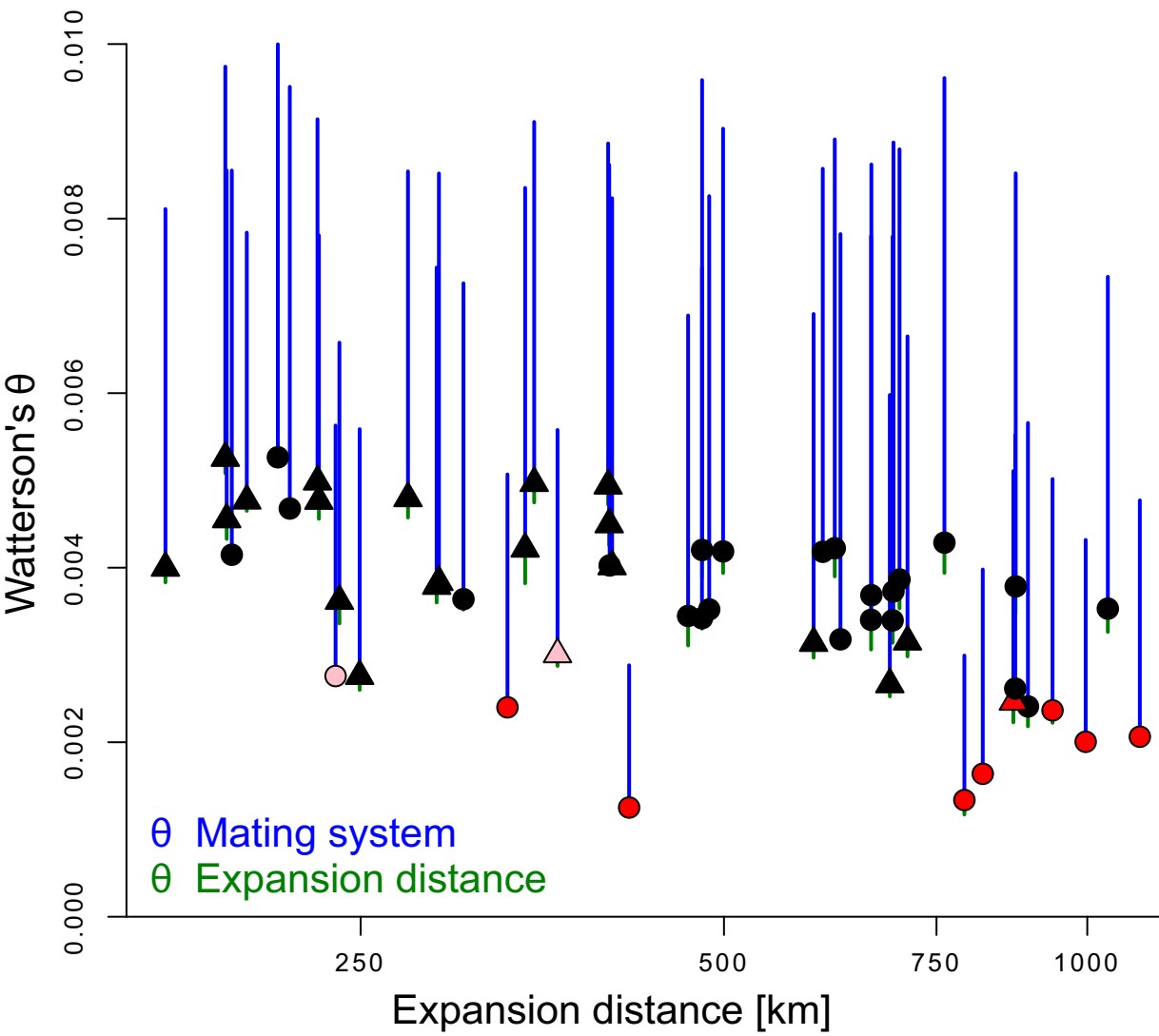

**Fig 5. Watterson's θ across all studied genic regions for each population against expansion distance (or distance from core for rear-edge populations, in log-scale).** Each symbol represents the average estimate of a population. The symbol type reflects the genetic cluster (circle for west, triangle for east) and the symbol colour the mating system: outcrossing (black), mixed-mating (pink), and selfing (red). Bars depict the difference in average genetic diversity between non-outlier genes and outlier genes linked to expansion distance (green) or mating system (blue).

significantly higher in predominantly selfing compared to predominantly outcrossing populations (Figs 1 and S4), and it extended further, potentially reflecting reduced effective recombination [27,80]. Consequently, the switch in mating system to selfing, even if it is only partial, can increase LD to a level that is equal or higher than otherwise observed only among outcrossing populations at the very edge of the geographic distribution (Figs 1 and S4).

Patterns of LD varied across the genome of *A. lyrata* (S2 Fig) and average genome-wide LD was significantly higher in genic compared to intergenic regions (Fig 3). The observation of increased LD in genic regions is consistent with findings in humans [81] and *A. thaliana* [82], and has generally been attributed to increased selection and/or reduced recombination within genes [83]. In *A. thaliana*, increased gene density was shown to be negatively correlated with genetic diversity, and increased LD seemed to result from increased background selection

and–to a lesser degree–from selective sweeps [46]. Consistent with these observations, we showed that LD in *A. lyrata* was negatively correlated with our proxies of gene density in the analyses of both genic and intergenic regions (Table 1). Because we found no support that background recombination was strongly linked with LD (S5 Fig), increased LD in genic regions of *A. lyrata* may be a result of selection–as in *A. thaliana*. Our results further suggest a complex interplay between demography and selection, as model-predicted increases in LD with range expansion and a shift in mating system to selfing were about five times stronger for genic compared to intergenic regions. A likely scenario is that range expansion and a shift to selfing increase background and directional selection on (combined) variants with larger phenotypic effects.

The increase in genome-wide LD with expansion and a shift in mating system to selfing involved neutral and deleterious polymorphisms. Focusing on annotated genic SNPs (Table 2, Fig 2), we showed here that the increase in LD with range expansion and mating system shift occurred to a similar degree when involving putatively neutral sites or a putatively neutral and a deleterious site (Fig 2). Indicated by the significant interaction terms in our model (Table 2), the increase in LD was less strong between two putatively deleterious sites (Fig 2). The latter could reflect the effect of increased purging of densely arranged deleterious mutations when they increase in frequency because of expansion and a mating system shift to selfing [84]. The overall pattern of LD being coupled with load was also confirmed by our population-level analyses, as average genome-wide LD for genic and intergenic regions was positively related with the number of deleterious counts (Fig 3B) but negatively with genetic diversity in intergenic regions, indicating higher LD in populations of small effective size [4,85] (Fig 3A). These results suggest that LD is a good predictor of mutational load caused by genetic drift [86], but similar studies in other systems are required to assess whether the pattern is general.

Postglacial range expansions are often limited in geographic extent–a pattern that is hypothesized to be driven by factors such as decreasing habitat suitability [79], loss of adaptive genetic variation during range expansion [87] and accumulation of mutational load [17]. Given our findings, we suggest that range expansions may also be limited by increasing LD because selection acting on one locus is likely to involve one or more deleterious loci that are linked. This is supported by theory, first by the Hill-Robertson inference, which postulates that increased LD reduces the efficacy of selection by reducing the probability for fixation because selection on one locus will interfere with selection on another linked locus [15,16]. While originally described for advantageous mutations [15], more recent theoretical [18] and empirical work [88,89] suggests that the Hill-Robertson inference may also apply when deleterious mutations are involved and may constrain adaptation [18]. Second, the model of Peischl et al. [17] proposed the scenario in which the expansion process leads to the heightened accumulation of mutational load if recombination is low, such that the two processes combined can be the sole reason for range limits. Several findings of this study suggest a combination of factors that restrict the distribution of *A. lyrata*: the strong increase in LD both with expansion distance and a shift to selfing particularly in genic regions, the involvement of deleterious SNPs in increased LD, and a general association between LD and deleterious counts. As declining drift-effective size towards range edges is a common pattern [90], and recombination is generally limited to some extent, the combined importance of load and LD to range limits might be of general relevance.

In addition to the genome wide-patterns of LD, we identified sets of genes for which the increase of LD was outside of the 95% confidence interval of genic LD to range expansion or shift in mating system. These genes have potentially experienced selection or are linked to genomic regions under selection caused directly by the two demographic processes or by environmental factors associated with them (S2 and S3 Tables). Along its range expansion, *A*.

*lyrata* has adapted to a variety of environments, and many of our outlier genes were indeed associated with traits known to vary along latitudinal clines or between sites with different temperature regimes and photoperiod lengths in *Arabidopsis* [76,91] and *A. lyrata* in particular [92,93]. For instance, the outlier genes for range expansion included *FPA*, which is involved in the autonomous pathway of flowering time regulation, as well as *PDP2*, which is part of the photoperiod-dependent pathway of flowering time regulation in *Arabidopsis* [76,91]. Similarly, the outlier genes for mating system shift included *AL7G32710*, the equivalent of *ARK3* in *A. thaliana*, flanking the region of the self-incompatibility locus. Increased LD across the self-incompatibility locus and its neighboring genes has previously been reported for *A. lyrata* [74,94]. Consistent with these findings, we found that both flanking genes showed increased LD among predominantly selfing populations (Fig 4). To which degree this pattern holds true for the two genes within the self-incompatibility locus that are responsible for self-incompatibility in *A. lyrata* could not be inferred because this genomic region is highly variable among populations [95] and requires individual sequencing of the target genes [96]. Heightened genetic diversity can be locally maintained by balancing selection, *e.g.* at the self-incompatibility locus [74,94,97] or in genomic regions coding for disease resistance [35]. Thus, the significantly increased level of overall genetic diversity that we observed in LD outlier genes for mating system (Fig 5) could reflect past balancing selection in some of these regions of the genome but requires further in-depth studies.

In summary, linkage disequilibrium, despite its widespread use and importance in genetics, is a conundrum [2]. Here we showed that genome-wide LD is strongly affected by both post-glacial range expansion and a shift in mating system toward selfing. Both processes increased LD substantially, for both presumably neutral and deleterious mutations, supporting that genome-wide LD is a good predictor of mutational load. The two demographic processes affected a few genomic regions even more strongly, potentially by imposing selection directly or by being associated with changes in the selection regime in response to a changing environment. Further theoretical explorations, ideally involving simulations, are though needed to establish a general framework for the impact and interplay of the factors driving LD during range expansion.

## Supporting information

**S1 Fig.** Decay of linkage disequilibrium (LD), measured as $r^2$ between SNPs, with increasing physical distance (bps) for genic (black) and intergenic (blue) regions for each population plotted separately (S1 Table). LD is based on the average across all retained pairwise LD estimates for each base-pair distance bin. Note that with increasing insert size, fewer observations occur, increasing the stochasticity in the plots.
(PDF)

**S2 Fig.** Average LD, measured as $r^2$ between SNPs from pool-seq data for each analyzed genic (black, grey) and intergenic (red, orange) region for each population across scaffolds I-VIII of the *A. lyrata* genome plotted for each population (S1 Table).
(PDF)

**S3 Fig. Distance in units of base pairs (bps) between consecutive genes along scaffolds 1–8 of the *Arabidopsis lyrata* genome.** The red lines indicate the position of the centromeres.
(EPS)

**S4 Fig.** Relationship between the average estimated LD based on pool-seq data and the correlation of zygosity (Δ) scaled by the level of genome-wide heterozygosity (θ) for genic (A) and intergenic (B) regions. Both estimates were calculated over the same genomic region, i.e. 500 bps. The symbol type reflects the genetic clusters (circle for west, triangle for east) and the

symbol color the mating system: outcrossing (black), mixed-mating (pink), and selfing (red).
(EPS)

**S5 Fig. Genome wide recombination across four populations from the range center estimated from pool-seq data.** A–recombination (c/bp) estimated for genic (black/grey) and intergenic (red/orange) regions used in this study with the blue lines indicating the smoothed average recombination across each genome. The eight scaffolds of the *A. lyrata* genome are indicated with alternating shading (black/red vs. grey/orange). Green bars indicate centromeric regions. B–relationship between recombination rate and LD estimated from pool-seq data for genic (top) or intergenic (bottom) regions.
(TIFF)

**S6 Fig.** Relationship between genome-wide linkage disequilibrium (LD, distance-corrected $r^2$ between pairs of SNPs) for genic (A) and intergenic (B) regions and expansion distance (or distance from core for rear-edge populations). Each symbol represents the average estimate of a population. The symbol type reflects the genetic clusters (circle for west, triangle for east) and the symbol color the mating system: outcrossing (black), mixed-mating (pink), and selfing (red).
(EPS)

**S7 Fig.** The level of significance (*p* values) adjusted for multiple testing with a false discovery rate (FDR) for the different fixed effects of linear mixed-effects models for genic (A-C) and intergenic (D-F) regions. Response variable was the correlation of zygosity ($\Delta$) scaled by the level of genome-wide heterozygosity ($\theta$) for each bp distance for 1–5,000 bps and in bins of 100 bps between 5'001–50'000 bps. *P* values that were significant (i.e. $p < 0.05$) after an FDR are highlighted in blue.
(TIFF)

**S1 Table. Overview of the 52 studied populations.** Indicated are the coordinates, the genetic cluster expansion distance from the core of each cluster, $F_{IS}$ based on microsatellite markers and the mean LD ($r^2$ corrected for distance between SNPs) for genic and intergenic regions for each population across the genome.
(XLSX)

**S2 Table. Genes that show a significant relationship between range expansion and LD following a FDR correction for multiple testing.** Given are the adjusted *p*-values and the GO terms associated with a particular gene. Where available the homolog of each gene in *A. thaliana* with the respective gene names and gene description is given. Genes and GO terms in bold are shared with the outlier genes for mating system. Colours highlight different GO trait categories or genes with known function for the respective category in *A. thaliana*.
(XLSX)

**S3 Table. Genes that show a significant relationship between mating system shifts and LD following a FDR correction for multiple testing.** Given are the adjusted *p*-values and the GO terms associated with a particular gene. Where available the homolog of each gene in *A. thaliana* with the respective gene names and gene description is given. Genes and GO terms in bold are shared with the outlier genes for range expansion. Genes highlighted in green have a known function related to growth in *A. thaliana*.
(XLSX)

**S1 Text. Results of the analyses for pool-seq data when omitting six populations that were old, i.e. established before the retreat of the glaciers following the last glaciation period.**
(DOCX)

## Acknowledgments

We thank Josh Van Buskirk for valuable comments on an earlier version of this manuscript and Mélissa Lemoine for statistical advice. Sequencing was done at the Genetic Diversity Centre, ETH Zürich, and at the Department of Biosystems Science and Engineering of ETH Zürich in Basel and the University of Basel. All analyses were performed at sciCORE (http://scicore.unibas.ch/) scientific computing center at the University of Basel.

## Author Contributions

**Conceptualization:** Kay Lucek, Yvonne Willi.

**Data curation:** Kay Lucek.

**Formal analysis:** Kay Lucek.

**Funding acquisition:** Yvonne Willi.

**Visualization:** Kay Lucek.

**Writing – original draft:** Kay Lucek.

**Writing – review & editing:** Kay Lucek, Yvonne Willi.

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
