## [Decision Letter · Decision Letter 0]

17 Jan 2021

Dear Dr Lucek,

Thank you very much for submitting your Research Article entitled 'Drivers of linkage disequilibrium across a species’ geographic range' to PLOS Genetics.

The manuscript was fully evaluated at the editorial level and by independent peer reviewers. The reviewers appreciated the attention to an important topic but identified some concerns that we ask you address in a revised manuscript

We therefore ask you to modify the manuscript according to the review recommendations. Your revisions should address the specific points made by each reviewer.

[LINK]

Yours sincerely,

Juliette de Meaux

Associate Editor

PLOS Genetics

Kirsten Bomblies

Section Editor: Evolution

PLOS Genetics

Dear Authors,

Your re-submission has now been revised by the same three reviewers who had initially looked at it. Two of them are satisfied with the changes, but there are still a number of comments voiced by one reviewer that you should incorporate. These revisions are minor but nevertheless substantial and you must address them thoroughly. I appreciate that you ended the discussion mentioning explicitly that, in the future, it will be interesting to model the two systems. Finally, I would like to suggest you refer to Takou et al. https://www.biorxiv.org/content/10.1101/709873v4, which provides a comparison of LD in an equilibrium and a bottlenecked population in the related subspecies A. lyrata ssp. petraea. This data is more exhaustive than ref. 31.

Please also revise Line 441: it seems to me that the factors that drive LD are known, but their relative importance in nature is seldom described.

Reviewer's Responses to Questions

**Comments to the Authors:**

Reviewer #1: The authors had seriously revised the manuscript following the reviewer and editor's comments. I think this is a timely and sound contribution to the filed of evolutionary ecology and population genetics. I don't have further comments on it.

Cheers,

Yongfeng

Dr. Yongfeng Zhou

Postdoctoral researcher @ The Gaut Lab

Department of Ecology and Evolutionary Biology

University of California, Irvine

Reviewer #2: The authors have addressed all the reviewers comments and the paper is undoubtedly better now. I am still a bit doubtful about the use of linear models as I feel they do not take into account the evolutionary variance of the process (that coming from the coalescent process). In this sense it may probably have been preferable to test the effects of mating system or range expansion through simulations.

Minor comments:

Introduction:

Page 22, 511-512: I am far from certain that LD is a good predictor of mutation laod and of a drift-syndrome (whatever the latter means).

Page 23: line 528-529: This conclusion seems rather strong.

Reviewer #3: First of all, I would like to apologize for my late answer.

In this revised version of the manuscript entitled “Drivers of linkage disequilibrium across a species’ geographic range” the authors made effort to answer to the comments and provided new analyses, including an estimate of the recombination map based on a new method that can be applied to poolseq data and the use of SIFT score to analyze the load.

I found that the framework of the manuscript was more clearly stated and some speculative parts were reduced.

Overall I found the manuscript very interesting and stimulating. However I still have some comments (mainly technical) that should be addressed before publication.

P4 L57: About the effect of bottleneck on LD, in Schaper et al. (2012) the authors not only showed that bottleneck can increase the extend of LD but that, under certain conditions at least, make LD less dependent to physical distant so generate LD at higher distance than expected by a simple Ne rescaling. I don’t think this prediction can be easily tested but at least it should be mentioned. Maybe one possibility would be to compare the observe pattern of decrease in LD with physical distance between SNPs with a simple neutral model. For ex on Fig 1C and 1D it is not clear how the curves have been obtained. What is the expected shape under neutrality and equilibrium? I’ve noted that the authors justify not too run complex simulations but at least comparison with neutral expectations is easily doable.

The recombination map is a nice addition but could have been exploited more.

- First some points should be precised. Was the map estimated using the four core populations pooled or separated? If the population structure is weak among them, they can be pooled otherwise it is better to perform distinct estimates. If distinct estimates have been obtained how are they averaged?

- The authors noted no correlation between LD and local recombination rate, which is a bit weird as recombination is based on LD information. A possible reason is that they only consider recombination estimate at a very small scale where precision can be low.

- It would be interesting to have a smoother view of recombination map (in addition to Fig S3). For example it seems that there are genomic regions (center of chromosomes?) with lower recombination rates.

- It is not clear also whether the correlation have been done only in the four core populations? What would be important to know is whether the increase in LD observed in peripheral and selfing populations is more pronounced in regions of low recombination (which can be categorized at a larger scale than done here).

They may have some issues with the linear model analysis. If I understood correctly a classical mixed linear model has been used. However, r2 is not a normal variable (0≤r2≤1) so at least a transformation is required. In addition, the mean r2 over 300 bp is used as the dependent variable. The precision on the estimate of this variable depends on the number of pairs of SNPs used to estimate it. Because this number increases with local diversity, the variance should also do it, which violate the heterosedasticity hypothesis. Linear model are usually quite robust to such violation but here it goes in the direction of the explanatory variables. This should be checked and alternative model should be used is necessary.

P13 L292-294: Instead of Fis, the RMES software could be used to get more accurate estimates of the selfing rates (David, P., B. Pujol, F. Viard, V. Castella, and J. Goudet. 2007. Reliable selfing rate estimates from imperfect population genetic data. Mol. Ecol. 16:2474-2487.).

P17 L383-386: Analysis of SIFT data. The authors found a (slight) difference between the LD between the different SIFT categories. However, they should have different frequencies, lower for DEL than TOL SNPs. This could explain the difference between the categories. This could be checked by redoing the analysis for different classes of frequencies. In Table 2, I don’t understand how a single effect can be obtained for the SIFT type given that there are three categories. This variable should be categorical with two values for the effect.

**Have all data underlying the figures and results presented in the manuscript been provided?**

Reviewer #1: Yes

Reviewer #2: Yes

Reviewer #3: Yes

PLOS authors have the option to publish the peer review history of their article (what does this mean?). If published, this will include your full peer review and any attached files.

Reviewer #1: No

Reviewer #2: No

Reviewer #3: No

---

## [Decision Letter · Decision Letter 1]

9 Mar 2021

Dear Dr Lucek,

We thank you for addressing all comments in this last round of reviews. After all this hard work, the good news: we are pleased to inform you that your manuscript entitled "Drivers of linkage disequilibrium across a species’ geographic range" has been editorially accepted for publication in PLOS Genetics. Congratulations!

Yours sincerely,

Juliette de Meaux

Associate Editor

PLOS Genetics

Kirsten Bomblies

Section Editor: Evolution

PLOS Genetics

Comments from the reviewers (if applicable):

Reviewer's Responses to Questions

**Comments to the Authors:**

Reviewer #3: The authors have carefully answered to my last comments and clarified the few points I've mentioned.

I have no additional comment and I think the manuscript is suitable for publication.

**Have all data underlying the figures and results presented in the manuscript been provided?**

Reviewer #3: Yes

PLOS authors have the option to publish the peer review history of their article (what does this mean?). If published, this will include your full peer review and any attached files.

Reviewer #3: No

**Data Deposition**

http://datadryad.org/submit?journalID=pgenetics&manu=PGENETICS-D-20-01727R1

**Press Queries**

---

## [Editor Report · Acceptance letter]

22 Mar 2021

PGENETICS-D-20-01727R1 

Drivers of linkage disequilibrium across a species’ geographic range 

Dear Dr Lucek, 

We are pleased to inform you that your manuscript entitled "Drivers of linkage disequilibrium across a species’ geographic range" has been formally accepted for publication in PLOS Genetics! Your manuscript is now with our production department and you will be notified of the publication date in due course.

With kind regards,

Alice Ellingham

PLOS Genetics

On behalf of:
